# Microbial profile in bile from pancreatic and extra-pancreatic biliary tract cancer

**Paola Di Carlo[1], Nicola Serra[2]\*, Teresa Maria Assunta Fasciana[1], Anna Giammanco[1], Francesco D'Arpa[3], Teresa Rea[2], Maria Santa Napolitano[1], Alessandro Lucchesi[4], Antonio Cascio[5], Consolato Maria Sergi[6]**

1 Department of Health Promotion, Maternal-Childhood, Internal Medicine of Excellence "G. D'Alessandro", University of Palermo, Palermo, Italy, 2 Department of Public Health, University Federico II of Naples, Naples, Italy, 3 Department of General Surgery and Emergency, University of Palermo, Palermo, Italy, 4 Hematology Unit, IRCCS Istituto Scientifico Romagnolo per lo Studio dei Tumori (IRST) "Dini Amadori", Meldola, Forl-Cesena, Italy, 5 Department of Health Promotion, Maternal-Childhood, Internal Medicine of Excellence "G. D'Alessandro", Infectious Disease Unit, University of Palermo, Palermo, Italy, 6 Lab. Med. and Pathology, Children's Hospital of Eastern Ontario (CHEO), Ottawa, Canada

\* nicola.serra@unina.it

## Abstract

### Background

Dysbiotic biliary bacterial profile is reported in cancer patients and is associated with survival and comorbidities, raising the question of its effect on the influence of anticancer drugs and, recently, the suggestion of perichemotherapy antibiotics in pancreatic cancer patients colonized by the *Escherichia coli* and *Klebsiella pneumoniae*.

### Objective

In this study, we investigated the microbial communities that colonize tumours and which bacteria could aid in diagnosing pancreatic and biliary cancer and managing bile-colonized patients.

### Methods

A retrospective study on positive bile cultures of 145 Italian patients who underwent cholangiopancreatography with PC and EPC cancer hospitalized from January 2006 to December 2020 in a QA-certified academic surgical unit were investigated for aerobic/facultative-anaerobic bacteria and fungal organisms.

### Results

We found that among Gram-negative bacteria, *Escherichia coli* and *Pseudomonas* spp were the most frequent in the EPC group, while *Escherichia coli*, *Klebsiella* spp, and *Pseudomonas* spp were the most frequent in the PC group. *Enterococcus* spp was the most frequent Gram-positive bacteria in both groups. Comparing the EPC and PC, we found a significant presence of patients with greater age in the PC compared to the EPC group. Regarding *Candida* spp, we found no significant but greater rate in the PC group compared

**Funding:** The author(s) received no specific funding for this work.

**Competing interests:** The author(s) have declared that no competing interests exist.

to the EPC group (11.7% vs 1.96%). We found that *Alcaligenes faecalis* was the most frequent bacteria in EPC than the PC group, among Gram-negative bacterial species.

## Conclusions

Age differences in gut microbiota composition may affect biliary habitats in our cancer population, especially in patients with pancreatic cancer. *Alcaligenes faecalis* isolated in the culture of bile samples could represent potential microbial markers for a restricted follow-up to early diagnosis of extra-pancreatic cancer. Finally, the prevalence of *Candida* spp in pancreatic cancer seems to trigger new aspects about debate about the role of fungal microbiota into their relationship with pancreatic cancer.

## Introduction

Among biliary tract and pancreatic malignancy disorders, extra-pancreatic biliary tract cancer, including biliary cholangiocarcinoma (CCA), gallbladder cancer (GBC), ampullary cancer (ACA), and pancreatic cancers, are rare. They considered aggressive malignancies and frequently fatal diseases [1–6]. However, the pancreatic death rate in Europe is increasing, and a cancer epidemiology survey of this emerging tumor showed a median survival time of < 6 months in 2018 [7, 8].

Studies on bile samples for microbiological analysis and to collect research data on the microbiota in patients with pancreas and biliary tract cancer are emerging [9–15]. Recently, a microbial community of the pancreas and the biliary tract cancers has been investigated to identify a distinct microbiota composition associated with cancer in the pancreatic and biliary habitats [11–15]. Moreover, some researchers are exploring how to change the pancreatic-biliary microbiota to slow tumor growth [16, 17].

Studies on bacterial bile composition have detected microorganisms with a possible role in carcinogenesis and chronic inflammation [18–24]. These findings encouraged the emergence of metagenomic, culture isolation studies and animal model studies in the most recent literature. [25–27].

Bile microbiota dysbiosis resulting in an imbalance in bacterial composition, changes in bacterial metabolic activities, or changes in the abundance of species in bacterial distribution, are involved in modifying host substrates such as bile acid components, which can result in the development of disease states including cancer cell proliferation [28–31].

Besides the well-known risk factors for pancreatic cancer, recently emerging studies have highlighted the role of human oral microbiota [32].

Our investigations on the culture of bile samples collected by endoscopic retrograde cholangiopancreatography (ERCP) have identified a pattern of bacteria isolated in patients with pancreatic-biliary diseases [20–23]. Dysbiotic biliary bacterial profile and cancer are associated with survival and comorbidities[ 22], raising the question of its effect on the influence of anti-cancer drugs and, recently, the suggestion of perichemotherapy antibiotics in pancreatic cancer patients colonized by *Klebsiella pneumoniae* [26, 31, 33].

Recent studies have shown that low-diversity microbiota and increased abundance of Enterococcus genus and other bacteria such as *Klebsiella pneumoniae* and *Escherichia coli* are observed in critically ill patients and are associated with a higher risk of infection [33–36]. If these microbiota changes have a critical role in severe infection in cancer patients is currently unknown and further studies are needed in this line of investigation [37, 38]. So, the culture

methodology is preferred for long-term strain storage and helps to verify that the same strains determine potential infection belonging to the microbiota [39, 40].

In this study, we report on a cohort of patients with malignant pathologies of the biliary and pancreatic tract to verify the microbiological signature of bile microbiota. In addition, we evaluate if there is a different microbiological pattern in subjects with pancreatic and extra-pancreatic biliary tract neoplasms.

## Materials and methods

### Methods

We retrospectively analyzed all positive surveillance bile cultures recovered from histopathologically confirmed patients with malignant biliary and pancreatic tumors in subjects who underwent chemotherapy treatment, according to cancer guidelines. Out of the patients with positive bile cultures, approximately 16% had either EPC or PC cancer. Patients with laboratory or clinical signs of sepsis using the definitions of the Third International Consensus for Sepsis and Septic Shock (Sepsis-3) were excluded from the study [41]. The patients who underwent ERCP were both subjects during the first surgery, and patients readmitted to our unit and out-patients emergency individuals from other hospitals of the Sicilian region (about 60% of endoscopic procedures per year for biliary and pancreatic malignancies). The commonly observed surgical procedures included pancreatic and biliary tract malignancies admitted to the "Paolo Giaccone" University Hospital in Palermo, Italy, from January 2006 to December 2020; data collected by medical records were archived in October 2021.

### Endoscopic techniques for biliary collection

During the ERCP procedure, a 5.5 Fr catheter with a single lumen was used to have a large lumen, reduce the timing of the procedure (biliary sampling), and minimize the risk of bile contamination. The wire-guide technique was used to avoid contamination. This procedure was used instead of contrast medium injection once access to the biliary tract was obtained. Biliary sampling was performed immediately after gaining access to the biliary tract before other endoscopic procedures, such as sphincterotomy, dilation, or stent placement. The ERCP procedure was performed following strictly aseptic endoscopic procedures operated at the Policlinico Paolo Giaccone University Hospital [42]. The University of Palermo Hospital is a quality assurance certified clinical institution with regular controls to acquire periodic certification. According to our institutional guidelines and standards, patients with valve defects or cardiac prostheses must receive antibiotic prophylaxis before undergoing an endoscopic procedure [42].

### Bile microbiological culture

The bile samples were inoculated for aerobic/facultative-anaerobic bacteria isolation on agar blood+ blood sheep 5%, McConkey agar, and Salmonella-Shigella agar media (Becton-Dickson). After 24h of incubation in aerobic conditions, the growth colonies were identified by matrix-assisted laser desorption ionization-time of flight mass spectrometry (MALDI-TOF MS) (MALDI Biotyper CA System, Bruker Daltonics Inc., USA) as previously published [43]. Microbiological records were systematically checked for the corresponding culture reports. Demographic data and diagnosis were collected from the patient's electronic medical history.

The epidemiological and clinical characteristics of the surgical setting were the same as previously reported [20]. This study was conducted among hospitalized adult patients using the International Classification of Diseases (ICD-10). The hospital setting is the Policlinic

University Hospital of Palermo, Italy. After ERCP, the enrolled patients were referred to a specialist cancer center.

To compare the isolates from the bile culture of patients with biliary cancer versus patients with pancreatic cancer, we stratified the patients into two groups, i.e., patients with extra-pancreatic biliary tract carcinoma (EPC) and pancreatic cancer (PC).

### Ethics statement

All procedures performed in studies involving human participants were in accordance with the ethical standards of the institutional and/or national research committee and with the 1964 Helsinki Declaration and its later amendments or comparable ethical standards. This study was approved by the Local Ethics Committee (AIFA code–CE 150109; protocol N° 09/2021).

No economic incentives were offered or provided for their participation.

### Statistical analysis

Data were presented as numbers or percentages for categorical variables. Continuous data are expressed as the mean ± standard deviation (SD) or median with Interquartile Range (IQR).

The chi-square and Fisher's exact tests were performed to evaluate significant differences in proportions or percentages between the two groups. Mainly, Fisher's exact test was used where the chi-square test was not appropriate. The multiple comparison chi-square tests were used to define significant differences among percentages. In this case, if the chi-square test was significant ($\alpha$ level: 0.05), the residual analysis with the Z-test was performed. Test for normal distribution was performed by Shapiro-Wilk test. The t-test was used to test the differences between two means of unpaired data. Alternative non-parametric tests were used when distribution was not normal. Notably, the Mann-Whitney test was used to compare two independent samples. All tests with *p-value(p)* $< 0.05$ were considered significant. The statistical analysis was performed by Matlab statistical toolbox version 2008 (MathWorks, Natick, MA, USA).

### Results

The positive bile samples were obtained from 145 consecutive patients, composed of 54.5% males and 45.5% females, with ages 49–94 years old, a mean age of 75.1 years old, and a standard deviation (SD) of 10. Especially, we considered in this study two groups of patients:

- EPC: composed of 51 patients with extra-pancreatic biliary tract carcinoma (cholangiocarcinoma (CCA) = 38 patients, gallbladder carcinoma (GBC) = 8 patients, ampullary carcinoma (ACA) = 5 patients); the group included 51% males and 49% females, with ages 62–93 years old, mean 75.5 years old and standard deviation of 8.

- PC: composed of 94 patients with 56.4% males and 43.6% females, with ages 49–94, mean of 76.1 years old, and SD of 10.5.

In **Table 1,** we reported the characteristics of 145 pancreatic and extra-pancreatic cancer patients. Gram-negative bacteria were isolated in 82.76% of bile cultures, while Gram-positive bacteria in 3.45% and Gram-positive and Gram-negative bacteria were detected in 11.06% of cancer patients. 12 out of 145 (8.28%) patients showed *Candida* spp in bile samples.

In **Table 2**, we report the isolates individuated in this study. Some patients showed more isolates (Gram-negative, Gram-positive, or both). Specifically, Table 2 indicates that among Gram-negative bacteria, the less frequent bacteria were *Morganella morganii*, *Pantoea agglomerans*, *Elizabethkingia meningoseptica*, *Serratia* spp, *Delftia acidovorans*, and *Brevundimonas* spp. In contrast, a significant most frequent rate was seen with *Klebsiella* spp, *E. coli*, and

**Table 1. Characteristics of 145 patients with pancreatic and extra-pancreatic biliary cancer.**

| Parameters | Sample |
|---|---|
| *Patients* | 145 |
| *Age at microbiological analysis* | |
| Mean±SD | 75.1±10 |
| Median (IQR) | 76 (69–82.25) |
| *Gender* | |
| Male | 54.5% (79) |
| Female | 45.5% (66) |
| *Cancer* | |
| Pancreatic | 64.8% (94) |
| Extra-pancreatic | 35.2% (51) |
| *Bacteria (# patients)* | 141 |
| *Gram -* | 82.76% (120) |
| *Gram+* | 3.45% (5) |
| *Both* | 11.03% (16) |
| *Fungus* (patients) | 12 |
| *Candida spp* only | 2.76% (4) |
| *Candida spp* and Gram- | 2.76% (4) |
| *Candida spp* and Gram+ | 2.76% (4) |
| *Candida spp* and Gram- and Gram+ | 0.0% (0) |

SD = standard deviation

IQR = Interquartile range

Gram— = Gram negative

Gram + = Gram positive.

*Pseudomonas* spp. In the Gram-positive bacteria, the less frequent strains were *Staphylococcus* spp and *Streptococcus* spp; while *Enterococcus* spp was the most frequent microorganism.

Table 3 reported the parameter percentages for patients in the PC and EPC groups. Table 3 shows that the PC group had an older patient than the EPC group (median: 79 vs 75, p = 0.0351). This result confirmed the significant relationship between age and cancer obtained in Table 3. In addition, there was more presence of fungus in the PC group than in the EPC group (11.7% vs 1.96%, p = 0.057), even if not significant.

Table 4 shows the isolates considering the EPC and PC groups. Particularly, in the EPC group, among Gram-negative bacteria, we observed a significantly less frequent rate for *Acinetobacter* spp; while *Escherichia coli* and *Pseudomonas* spp were the most frequent microorganisms. In PC group, the Gram-negative bacteria significantly less frequent were *Morganella morganii*, *Pantoea agglomerans*, *Elizabethkingia meningoseptica*, *Serratia* spp, *Brevundimonas* spp, *Alcaligenes faecalis*, and *Enterobacter* spp. In contrast, *Escherichia coli*, *Klebsiella* spp, and *Pseudomonas* spp were the most frequent Gram-negative bacteria. In addition, about Gram-positive bacteria, both groups had *Enterococcus* spp, the most frequent bacteria.

Comparing the PC and EPC groups, we found no significant difference for total Gram-negative and Gram-positive bacteria (81.91% vs 84.31%, p = 0.72; 13.83% vs 15.69%, p = 0.76, respectively). However, a significant association was observed between cancer groups and Gram-negative bacterial species (p = 0.0073). By post hoc z-test, we found that *Alcaligenes faecalis* was the most frequent bacteria in the EPC group (p = 0.039), while *Acinetobacter* spp was the bacteria less frequent(p = 0.014). In contrast, in the PC group, we found only significantly less frequent bacteria, such as *Morganella morganii* (p = 0.0196), *Pantoea agglomerans*

**Table 2. Strains isolated from 145 patients with pancreatic and extra-pancreatic cancer.**

| Isolated Strains | % (Number of Patients) |
|---|---|
| *Gram- (Patients)* | 82.76% (120)† |
| *Morganella morganii* | 0.83% (1)*** |
| *Pantoea agglomerans* | 0.83% (1)*** |
| *Elizabethkingia meningoseptica* | 1.67% (2)*** |
| *Serratia* spp | 1.67% (2)*** |
| *Delftia acidovorans* | 2.50% (3)*** |
| *Brevundimonas spp* | 2.50% (3)*** |
| *Alcaligenes faecalis* | 3.33% (4) |
| *Enterobacter* spp | 4.17% (5) |
| *Achromobacter* spp | 5.83% (7) |
| *Citrobacter* spp | 5.83% (7) |
| *Acinetobacter* spp | 6.67% (8) |
| *Stenotrophomonas* spp | 8.33% (10) |
| *Klebsiella* spp | 15.0% (18)** |
| *Escherichia coli* | 24.17% (29)** |
| *Pseudomonas* spp | 31.67% (38)** |
| *Gram+ (Patients)* | 14.48% (21)† |
| *Staphylococcus* spp | 4.76% (1)*** |
| *Streptococcus* spp | 9.52% (2)*** |
| *Enterococcus* spp | 90.48% (19)** |
| *Candida* spp | 8.28% (12) |

† = there are patients with more different isolates

** = significant most frequent isolate

*** = significant less frequent isolate.

(p = 0.0196), *Serratia* spp (p = 0.0104), *Alcaligenes faecalis* (p = 0.002), and *Enterobacter* spp (p = 0.0159). About the Gram-positive bacterial species, no significant relationship between PC and EPC group was found (p = 0.69).

Finally, in S1 and S2 Tables, we reported our stratified database into EPC and PC groups.

## Discussion

In this study, we investigated through bile culture which members of the bile microbiota could aid in diagnosing Pancreatic and Biliary Cancer. We found a prevalence of Gram-negative Enterobacteriaceae in pancreaticobiliary cancers. Comparing pancreatic and extra-pancreatic malignancy disease, we found that *Alcaligenes faecalis* was more correlated to extra-pancreatic cancers (EPC). Conversely, bacteria such as *Morganella morganii and Pantoea agglomerans* that have been shown to prevent excessive bacterial translocation and inflammation were significantly less frequent in pancreatic cancer patients [15,30].

For elderly Europeans, extra-pancreatic and pancreatic cancer incidence increased with age, and survival decreased [6–8]. Compared to other European countries, Italy showed a higher percentage of older people [44, 45].

In our study, the EPC group mainly included cholangiocarcinoma patients; the average age at diagnosis for people with cholangiocarcinoma is about 70 years, and the average age of pancreatic cancer is greater than 70 years.

**Table 3. General characteristics and comparison between EPC and PC group.**

| Parameters | EPC Group | PC Group | EPC vs. PC |
|---|---|---|---|
| | | | p-value (Test) |
| *Patients* | 51 | 94 | |
| *Age at microbiological analysis* | | | **0.0351*(MW)** |
| Mean±SD | 73.37 ± 8.84 | 76.05 ± 10.5 | |
| Median (IQR) | 75 (68–78.75) | 79 (70–83) | |
| *Gender* | | | 0.53 (C) |
| Male | 51.0% (26) | 56.38% (53) | |
| Female | 49.0% (25) | 43.62% (41) | |
| *Bacteria* | | | 1.0 (F) |
| Gram- | 84.3% (43) | 81.91% (77) | |
| Gram+ | 3.92% (2) | 3.19% (3) | |
| Both | 11.76% (6) | 10.64% (10) | |
| *Candida spp (patients)* | 1.96% (1/51) | 11.7% (11/94) | 0.057 (F) |
| *Candida* spp only | 0.0% (0) | 11.7% (4) | 1.0 (C) |
| *Candida* spp and Gram- | 1.96% (1) | 3.19% (3) | |
| *Candida* spp and Gram+ | 0.0% (0) | 0.0% (0) | |
| *Candida* spp and Gram- and Gram+ | 0.0% (0) | 11.7% (4) | |

* = significant test

MW = Mann Whitney test was used in the case the distribution was not normal

C = Chi-square test

F = Fisher's exact test.

Our study showed a mean and median age of pancreatic cancer patients older than extra-pancreatic biliary cancer subjects. Our pancreatic cancer subjects were elderly patients aged more than 75 years.

This result could be linked to the Italian median age and late diagnosis age following the European countries where pancreatic cancer is reported in subjects with a mean age of 70–80 [44].

Many intestinal bacteria can translocate to the liver, particularly in malignant tumours of the biliary system, and fungi and bacteria can invade the bile [45–49].

In pancreatic and extra-pancreatic biliary tract cancer patients, we found a higher prevalence rate of gut species of Gram-negative *Enterobacteriaceae*. They are well-known in gut dysbiosis, such as *Escherichia coli* and *Pseudomonas* spp prevalent in both groups, while *Klebsiella* spp was more frequent only in the PC group.

Moreover, in our study, all patients with pancreatic and extra-pancreatic cancer had a significant presence of *Pseudomonas* spp in the bile culture. In contrast, *Acinetobacter* spp was significantly less present in extra-pancreatic cancer than in the pancreatic cancer group.

These findings may be interesting because the researchers found distinctive microbial communities in a highly aggressive pancreatic tumor subtype called 'basal-like' and identified an increasing abundance of the *Pseudomonas* genus [50]. Since a significant presence of *Pseudomonas* spp in extra-pancreatic and pancreatic groups was observed in our study, the authors suggest further investigations on the role of *Pseudomonas* spp on aggressive tumors of biliary tract.

Regarding *Escherichia coli*, we found a significant presence of this bacteria in the bile samples of pancreatic and extra-pancreatic patients. This colonic contaminant bacterium is part of gut microbial dysbiosis associated with hepatocellular carcinoma via the gut-liver axis. *Escherichia coli* is under investigation as an emerging pathogen for pancreatic cancer progression [51].

**Table 4. Bacteria species isolated from EPC and PC group.**

| Isolates on 145 patients | EPC Group | PC Group | EPC vs. PC |
|---|---|---|---|
| | % (N = 51) | % (N = 94) | p-value (Test) |
| *Gram- (Patients)* | 84.31% (43/51) † | 81.91% (77/94) † | 0.72 (C) |
| *Morganella morganii* | 1.96% (1) | 0.0% (0)*** | |
| *Pantoea agglomerans* | 1.96% (1) | 0.0% (0)*** | |
| *Elizabethkingia meningoseptica* | 1.96% (1) | 1.06% (1)*** | |
| *Serratia* spp | 3.92% (2) | 0.0% (0)*** | |
| *Delftia acidovorans* | 1.96% (1) | 2.13% (2) | **0.0073* (F)** |
| *Brevundimonas* spp | 3.92% (2) | 1.06% (1)*** | *Alcaligenes faecalis (EPC)**, p = 0.039(Z)* |
| *Aalcaligenes faecalis* | 7.84% (4) | 0.0% (0)*** | *Acinetobacter spp (EPC)***, p = 0.014(Z)* |
| *Enterobacter* spp | 7.84% (4) | 1.06% (1)*** | *Morganella morganii (PC)***, p = 0.0196(Z)* |
| *Acinetobacter* spp | 0.0% (0)*** | 8.51% (8) | *Pantoea agglomerans (PC) ***, p = 0.0196(Z)* |
| *Achromobacter* spp | 5.88% (3) | 4.23% (4) | *Serratia spp (PC) ***, p = 0.0104(Z)* |
| *Citrobacter* spp | 3.92% (2) | 5.32% (5) | *Alcaligenes faecalis (PC)***, p = 0.002(Z)* |
| *Stenotrophomonas* spp | 3.92% (2) | 8.51% (8) | *Enterobacter spp (PC) ***, p = 0.0159(Z)* |
| *Klebsiella* spp | 9.80% (5) | 13.03% (13) ** | |
| *Escherichia coli* | 23.53% (12)** | 18.09% (17)** | |
| *Pseudomonas* spp | 21.57% (11)** | 28.72% (27)** | |
| *Gram+ (Patients)* | 15.69% (8/51) | 13.83% (13/94) † | 0.76 (C) |
| *Staphylococcus* spp | 0.0% (0) | 7.69% (1) | 0.69(F) |
| *Streptococcus* spp | 0.0% (0) | 15.38% (2) | |
| *Enterococcus* spp | 100% (8)** | 84.62% (11)** | |

† = there are patients with more different isolates

* = significant test

** = significant most frequent

*** = significant less frequent

C = chi-square test

F = Fisher's exact test

Z = post hoc z-test.

*Escherichia coli*, *Klebsiella* spp, and *Pseudomonas* spp isolated in our bile samples are known as Multi Drug-Resistant (MDR) pathogens responsible for hospital bloodstream infection in the Mediterranean basin area [52, 53].

Regarding *Escherichia coli*, and *Klebsiella pneumoniae*, recent investigations focused on these strains about the survival of pancreatic cancer patients with biliary colonization at the time when they underwent anti-cancer treatment [24, 33].

The authors suggest taking note of bile microbial composition and considering antibiotic resistance, as previously reported by Di Carlo et al. [21], where *E. coli*, *K. pneumonia*, and *P. aeruginosa* showed a high percentage of resistance to third-generation cephalosporin (3GC), aminoglycosides class and quinolone group in cancer patients in comparison to subjects with benign biliary disorders such as choledocholithiasis.

*Alcaligenes faecalis* was significantly found in EPC groups. This bacterium was considered an extensively drug-resistant pathogen responsible for bacteremia and sepsis in cancer patients with severe neutropenia [54, 55]. Recently, in a mouse model, *Alcaligenes faecalis* has been reported as lymphoid-tissue–resident commensal bacteria in intestinal lymphoid tissue (LRCs) that may modulate the host immune system and are involved in chronic intestinal inflammation [56]. Since the high concentrations of cholic acid inhibit the growth of

Alcaligenes, the alterations of bile metabolites during extra-pancreatic cancer play a role in the colonization of bile in the EPC group [57].

*Enterococcus spp* was the most frequent isolate among Gram-positive bacteria in pancreatic than extra-pancreatic, biliary patients. Maekawa *et al.* [58] showed as *Enterobacter* species and *Enterococcus faecalis* were significant components of the bacterial populations in the bile of patients with chronic pancreatitis and pancreatic cancer. Our study found *Enterobacter* spp less frequently in patients with pancreatic cancer bile samples. In contrast, *Enterococcus* spp was commonly isolated in the bile sample of pancreatic and extra-pancreatic biliary cancer subjects. Although gut bacteria can invade bile, the mouse model suggests this happens through Vater's ampulla for *Enterococcus spp*. While it is unknown whether *Enterococcus* spp infection directly causes pancreatic cancer or exacerbates an existing condition, emerging studies suggest that *Enterococcus* spp may play an oncogenetic role in liver cancer in subjects with chronic liver disease [59, 60].

As mentioned above, oral microbiota has been considered a biomarker in pancreatic cancer [32, 61].

In our pancreatic studied population, we found certain subgingival non-oral species, most notably *Acinetobacter baumannii* and *Pseudomonas aeruginosa*, reported in immunocompromised patients [62].

In our sample the almost significant of *Candida* spp presence in subjects with pancreatic cancer than in subjects with extra-pancreatic biliary tract carcinoma (p = 0.057), should be observed very carefully. In fact, the authors invite to perform an investigation on a larger sample to verify if this result could be significant. In this case, in fact, the *Candida* spp presence in the PC group could be an element which, combined with some specific bacteria, could be a marker of pancreatic cancer.

Fungi can migrate from the gut lumen to the pancreas, and the role of gut mycobiome in pancreas tumorigenesis is under debate. Its involvement remains unknown mainly and, in our opinion and of other authors, not satisfactorily investigated [14, 62–65]. Matsukawa et al. have recently investigated correlation networks targeting the combined oral and gut dysbiosis observed in pancreatic cancer patients [61].

In conclusion, different bile microbiota signatures are found in biliary and pancreatic cancer. The authors would like to stress the role of the signature of bile microbiota in rare and aggressive cancers like pancreatic cancer. There is mounting evidence supporting the part of the microbiome in response to cancer therapy. Therefore, vigorous efforts may be needed to study the intricate interplay of perturbations of the bile microbiota composition in patients with biliary and pancreatic cancer. Novel strategies in improving biliary tract and pancreatic diseases via modulating the gut microbiome are incoming [31, 66, 67].

## Limits of the study

In this research study, there was no control group of healthy subjects because there are ethical concerns [68, 69] related to the risks associated with the ERCP procedure [70]. In addition, a recent study by D'Amico at al., showed that the bile samples studied by rDNA sequence analysis of multiorgan donors who have not undergone any previous biliary procedures and have no biliary abnormalities is sterile [71].

In our study, there was no control group for benign pancreatic-biliary disorders and antibiotic profile of species isolated, because of ethical issues.

This study involved patients with pancreatic and extra-pancreatic cancers from 2006 to 2020, who were receiving chemotherapy according to standards and guidelines strictly adherent to international oncological protocols. In this period both the management of cancer

patients and the chemotherapy treatments, are changed, therefore the authors could not verify the impact of different chemotherapy treatments on patients' microbial profile. This aspect will be part of a subsequent investigation.

## Supporting information

**S1 Table. Gram-negative, positive and both isolated composition in PC group.** Particularly, we reported for each subgroup the patients with fungus presence, and in the last row the patients with fungus only.
(DOCX)

**S2 Table. Gram-negative, positive and both isolated composition in EPC group.** Particularly, we reported for each subgroup the patients with fungus presence, and in the last row the patients with fungus only.
(DOCX)

## Author Contributions

**Conceptualization:** Paola Di Carlo, Nicola Serra.

**Data curation:** Teresa Rea, Maria Santa Napolitano, Alessandro Lucchesi.

**Formal analysis:** Nicola Serra.

**Investigation:** Teresa Maria Assunta Fasciana, Anna Giammanco, Francesco D'Arpa.

**Methodology:** Teresa Maria Assunta Fasciana, Francesco D'Arpa.

**Supervision:** Antonio Cascio, Consolato Maria Sergi.

**Writing – original draft:** Paola Di Carlo, Nicola Serra, Consolato Maria Sergi.

**Writing – review & editing:** Paola Di Carlo, Anna Giammanco, Antonio Cascio.

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
