## [Decision Letter · Decision Letter 0]

26 Jul 2023

PONE-D-23-14947Microbial profile in bile from Pancreatic and Extra-Pancreatic Biliary Tract CancerPLOS ONE

Dear Dr. Serra,

Thank you for submitting your manuscript to PLOS ONE. After careful consideration, we feel that it has merit but does not fully meet PLOS ONE’s publication criteria as it currently stands. Therefore, we invite you to submit a revised version of the manuscript that addresses the points raised during the review process.

We look forward to receiving your revised manuscript.

Kind regards,

Massimiliano Galdiero, M.D., Ph.D.

Academic Editor

PLOS ONE

Journal Requirements:

Reviewers' comments:

Reviewer's Responses to Questions

**Comments to the Author**

1. Is the manuscript technically sound, and do the data support the conclusions?

Reviewer #1: Yes

Reviewer #2: Yes

2. Has the statistical analysis been performed appropriately and rigorously? 

Reviewer #1: Yes

Reviewer #2: Yes

3. Have the authors made all data underlying the findings in their manuscript fully available?

Reviewer #1: Yes

Reviewer #2: Yes

4. Is the manuscript presented in an intelligible fashion and written in standard English?

Reviewer #1: Yes

Reviewer #2: No

5. Review Comments to the Author

Reviewer #1: In this work, the authors analyzed the microbial communities that colonize tumors and focused on 'marker' bacteria in the diagnosis of pancreatic and biliary tract cancer and in the management of bile-colonized patients. The work is well written. The results are clear and add more information to the growing literature on microbiome associated with pancreatic and biliary neoplasia. However, I have minor comments to improve the text:

1) The bile samples were collected during the ERCP procedure. In following paper doi: 10.3389/fsurg.2021.621525 is reported that “….Collections of the bile samples were done following a strict protocol to ensure aseptic conditions, avoiding any possible microbial contamination. Bile was aspirated with a sterile 10-ml syringe. Five to ten milliliters of bile were enough for each sample”. Can you add your bile collection protocol?

2) Have you used a group of control patient to evaluate if these bacteria were mostly absent in healthy samples? Can you add their informations to the list of enrolled patients?

3) Your study was carried out on "...hospitalized adult patients", surely most part of them were treated with antineoplastic therapies. Can you report in “Material and method section” a sentence to clarify that your enrolled patients were treated or not treated? It is report that different chemotherapic treatments could influence patient microbial profile.

Reviewer #2: The authors conducted a retrospective study of bile samples from 145 patients with pancreatic or extrapancreatic cancer who underwent cholangiopancreatography between January 2006 and December 2020.

The study is interesting due to the number of patients enrolled and the still poor knowledge on this topic. However, I have some questions and comments for the Authors.

Materials and methods

The authors analyzed the results of culture performed on bile samples. The isolates are all aerobic or facultative-aerobic bacteria. Please describe the media and incubation conditions used: were they suitable for the recovery of strict anaerobes?

The authors investigated which members of the biliary microbiota could aid in the diagnosis of pancreatic and biliary cancer. It would be helpful to indicate what percentage of PC or EPC patients had culture-positive bile samples. Wasn't a control group available such as patients with benign pancreatico-biliary disorders?

Had the patients taken preoperative antibiotics? Preoperative prophylaxis would have influenced the results, but no patient exclusion criteria were indicated. Please add.

Line 116. The MALDI-ToF method is not described in the indicated reference. Please check.

Line 122. Please replace with: patients with extra-pancreatic biliary tract carcinoma (EPC).......etc.

Results

Line 151. Please replace with: 51 patients with extra-pancreatic biliary tract carcinoma (etc..)

Discussion

Line 247. The authors suggest taking note of the microbial composition of the bile and considering antibiotic resistance. Have their patients had post-operative infections? based on antibiotic susceptibility profiles, are their E. coli and K. pneumoniae isolates MDR?

Line 270. The frequency of Candida isolation is not significantly different between PC and EPC patients. The results obtained do not allow to hypothesize the correlation between periodontal disease and CP. Please rephrase.

Please check that all family, genus and species names are in italics.

Line 161: replace "candida" with Candida".

English needs to be revised in some points: lines 56, 171, 177, 185, 219, 227, 244.

6. PLOS authors have the option to publish the peer review history of their article (what does this mean?). If published, this will include your full peer review and any attached files.

Reviewer #1: No

Reviewer #2: No

---

## [Author Response · Author response to Decision Letter 0]

29 Aug 2023

Reviewer #1: 

In this work, the authors analyzed the microbial communities that colonize tumors and focused on 'marker' bacteria in the diagnosis of pancreatic and biliary tract cancer and in the management of bile-colonized patients. The work is well written. The results are clear and add more information to the growing literature on microbiome associated with pancreatic and biliary neoplasia. However, I have minor comments to improve the text:

[Reviewer 1]: 1) The bile samples were collected during the ERCP procedure. In following paper doi: 10.3389/fsurg.2021.621525 is reported that “….Collections of the bile samples were done following a strict protocol to ensure aseptic conditions, avoiding any possible microbial contamination. Bile was aspirated with a sterile 10-ml syringe. Five to ten milliliters of bile were enough for each sample”. Can you add your bile collection protocol?

[Authors]: Thank you for your suggestion. the authors in Materials and Methods cited the Guideline for Endoscopic Procedures, performed at Policlinico Paolo Giaccone Hospital, prot. nr, 97, 2016

(https://intranet.policlinico.pa.it/pub/documenti/browse.do?dispatch=documentRead&documentId=1323c0545872ac910158731923c50019).

Moreover, the following paragraph was added 

Endoscopic Techniques for Biliary Collection

During the ERCP procedure, a 5.5 Fr catheter with a single lumen was used to have a large lumen of vision, to reduce the timing of biliary sampling, and minimize the risk of bile contamination. The wire-guide technique was used to avoid contamination instead of contrast medium injection once access to the biliary tract was obtained. Biliary sampling is performed immediately after gaining access to the biliary tract before other endoscopic procedures, such as sphincterotomy, dilation, or stent placement. The ERCP procedure was performed following the aseptic endoscopic procedures of the Policlinico Paolo Giaccone University Hospital �42�. According to our local guidelines, patients with valve defects or cardiac prostheses must receive antibiotic prophylaxis before undergoing an endoscopic procedure �42�.

[Reviewer]: 2) Have you used a group of control patient to evaluate if these bacteria were mostly absent in healthy samples? Can you add their informations to 

the list of enrolled patients?

[Reply]: Thank you for your suggestion. The authors did not study the bile bacterial composition in healthy samples because the ERCP procedure is limited due to ethical concerns associated with complications post-ERCP procedures. Currently, diagnostic ERCP is not commonly used and has been replaced by CT, MRCP, and EUS imaging for diagnostic purposes. However, ERCP is still a crucial procedure for treating choledocholithiasis, bile duct leak, and relieving malignant obstructive jaundice.

The following sentences and references were added in the limitation section: 

Research on the microbial composition of healthy individuals is limited due to ethical concerns related to the risks associated with the ERCP procedure. ERCP, despite efforts to lessen complications, is still an invasive endoscopic procedure that continues to be extensively utilized for the treatment of choledocholithiasis, bile duct leakage, and the relief of malignant obstructive jaundice. Moreover, a recent study by D’amico and colleagues found that the bile samples studied by rDNA sequence analysis of multiorgan donors who have not undergone any previous biliary procedures and have no biliary abnormalities is sterile 

[68] World Health Organization. "WHO guidelines on ethical issues in public health surveillance." (2017).

[69] Marrone, M., Macorano, E., Lippolis, G., Caricato, P., Cazzato, G., Oliva, A., & De Luca, B. P. (2023). Consent and Complications in Health Care: The Italian Context. Healthcare, 11(3). https://doi.org/10.3390/healthcare11030360

[70] Solomon, S., & Baillie, J. (2019). Indications for and Contraindications to ERCP. ERCP (Third Edition), 54-58.e1. https://doi.org/10.1016/B978-0-323-48109-0.00007-9

[71] D'Amico F, Bertacco A, Finotti M, Di Renzo C, Rodriguez-Davalos MI, Gondolesi GE, Cillo U, Mulligan D, Geibel J. Bile Microbiota in Liver Transplantation: Proof of Concept Using Gene Amplification in a Heterogeneous Clinical Scenario. Front Surg. 2021 Mar 16;8:621525. doi: 10.3389/fsurg.2021.621525. PMID: 33796547; PMCID: PMC8009296.

[Reviewer]: 3) Your study was carried out on "...hospitalized adult patients", surely most part of them were treated with antineoplastic therapies. Can you report in “Material and method section” a sentence to clarify that your enrolled patients were treated or not treated? 

[Reply]: Thank you for your suggestion. We reported in Materials and Methods the sentence:

in subjects underwent to chemotherapy treatment, according to cancer guidelines.

[Reviewer]: It is report that different chemotherapic treatments could influence patient microbial profile.

[Reply]: Biliary sampling was performed considering this possible methodological bias. Therefore, the "materials and methods" we enrolled all patients undergoing chemotherapy.

Regarding the different types of chemotherapy used to treat oncological diseases of the biliary tract and pancreas, it is known that different chemotherapy treatments can lead to a different biliary microbiological profile of the patient. In this regard, it must be considered that this study involves patients over up ten years, during which the therapies have changed. We added in the Limitations paragraph the following sentence:

This study involved patients with pancreatic and extra-pancreatic cancers from 2006 to 2020, who were receiving chemotherapy according to standards and guidelines strictly adherent to international oncologcical protocols. In this period both the management of cancer patients and the chemotherapy treatments, are changed, therefore the authors could not verify the impact of different chemotherapy treatments on patients’ microbial profile. This aspect will be part of a subsequent investigation. 

Reviewer #2: 

The authors conducted a retrospective study of bile samples. from 145 patients with pancreatic or extrapancreatic cancer who underwent cholangiopancreatography between January 2006 and December 2020. The study is interesting due to the number of patients enrolled and the still poor knowledge on this topic. However, I have some questions and comments for the Authors.

[Reviewer]: Materials and methods. The authors analysed the results of culture performed on bile samples. The isolates are all aerobic or facultative-aerobic bacteria. Please describe the media and incubation conditions used: were they suitable for the recovery of strict anaerobes? 

[Reply]: Thank you for your comment. The authors have added the following subparagraph in the Materials and Methods and the new reference 43 and changed the abstract

Bile microbiological culture

The bile samples were inoculated for aerobic/facultative-anaerobic bacteria isolation on agar blood+ blood sheep 5%, McConkey agar, and Salmonella-Shigella agar media (Becton-Dickson). After 24h of incubation in aerobic conditions, the growth colonies were identified by matrix-assisted laser desorption ionization-time of flight mass spectrometry (MALDI-TOF MS) (MALDI Biotyper CA System, Bruker Daltonics Inc., USA) as previously published [43]. Microbiological records were systematically checked for the corresponding culture reports. Demographic data and diagnosis were collected from the patient's electronic medical history. 

The epidemiological and clinical characteristics of the surgical setting were the same as previously reported [20]. This study was conducted among hospitalized adult patients using the International Classification of Diseases (ICD-10). The hospital setting is the Policlinic University Hospital of Palermo, Italy. After ERCP, the enrolled patients were referred to a specialist cancer center. 

To compare the isolates from the bile culture of patients with biliary cancer versus patients with pancreatic cancer, we stratified the patients into two groups, i.e., patients with extra-pancreatic biliary tract carcinoma (EPC) and pancreatic cancer (PC). 

[Reviewer]: The authors investigated which members of the biliary microbiota could aid in the diagnosis of pancreatic and biliary cancer. It would be helpful to indicate what percentage of PC or EPC patients had culture-positive bile samples. 

[Reply]: Thank you for your suggestion. We added the sentence in the Methods at lines 107-108 (clean file):

“Out of the patients with positive bile cultures, approximately 16% had either EPC or PC cancer.”

[Reviewer]: Wasn't a control group available such as patients with benign pancreaticobiliary disorders?

[Reply]: Thank you for your suggestion. The authors have added the following sentences in the limit of the study. 

In our study, there was no control group for benign pancreatic-biliary disorders and antibiotic profile of species isolated, because of ethical issues. 

[Reviewer]: Had the patients taken preoperative antibiotics? Preoperative prophylaxis would have influenced the results, but no patient exclusion criteria were indicated. Please add.

[Reply]: thank you for your question. We added the sentence in subparagraph Endoscopic Techniques for Biliary Collection at line 127-129 (clean file)

According to our institutional guidelines and standards, patients with valve defects or cardiac prostheses must receive antibiotic prophylaxis before undergoing an endoscopic procedure �42�.

Guideline for Endoscopic Procedures, Policlinico Paolo Giaccone Hospital, prot. nr, 97, 2016. 

https://intranet.policlinico.pa.it/pub/documenti/browse.do?dispatch=documentRead&documentId=1323c0545872ac910158731923c50019

[Reviewer]: Line 116. The MALDI-ToF method is not described in the indicated reference. Please check.

[Reply]: We change the reference with this indicated following 

• Neelja Singhal, Manish Kumar, Pawan K. Kanaujia, and Jugsharan S. Virdi. MALDI-TOF mass spectrometry: an emerging technology for microbial identification and diagnosis. Front Microbiol. 2015; 6: 791. Published online 2015 Aug 5. doi: 10.3389/fmicb.2015.00791

[Reviewer]: Line 122. Please replace with: patients with extra-pancreatic biliary tract carcinoma (EPC).......etc.

[Reply]: Thank you for your helpful suggestion. The sentence has been changed.

[Reviewer]: Results - Line 151. Please replace with: 51 patients with extra-pancreatic biliary tract carcinoma (etc..)

[Reply]: Thank you for your helpful suggestion. The sentence has been changed.

[Reviewer]: Discussion - Line 247. The authors suggest taking note of the microbial composition of the bile and considering antibiotic resistance. Have their patients had post-operative infections? 

[Reply]: Thank you for your question. Regarding post-ERPC infection, in our sample, 60% of patients were lost to follow-up, while the remaining 40% showed no post-ERPC infection after seven days; where in 7% of cases (4 patients), we found nosocomial infection within 30 days by K. pneumonie.

[Reviewer]: Discussion - Line 247. Based on antibiotic susceptibility profiles, are their E. coli and K. pneumoniae isolates MDR?

[Reply]: Thank you for your question. In this study we not investigate on the antibiotic susceptibility profiles. In our previous study 

• [Di Carlo P, Serra N, D'Arpa F, Agrusa A, Gulotta G, Fasciana T, et al. The microbiota of the bilio-pancreatic system: a cohort, STROBE-compliant study. Infect Drug Resist. 2019;12:1513-27. DOI 10.2147/IDR.S200378] 

we reported that E. coli, K. pneumonia, and P. aeruginosa showed a high percentage of resistance to third-generation cephalosporin (3GC), aminoglycosides class and quinolone group in cancer patients in comparison to subjects with benign biliary disorders such as choledocholithiasis [lines 271-274].

[Reviewer]: Line 270. The frequency of Candida isolation is not significantly different between PC and EPC patients. The results obtained do not allow to hypothesize the correlation between periodontal disease and CP. Please rephrase.

[Reply]: Thank you for your suggestions. the following sentences were changed and references added. 

“In our sample the almost significant of Candida spp presence in subjects with pancreatic cancer than in subjects with extra-pancreatic biliary tract carcinoma (p= 0.057), should be observed very carefully. In fact, the authors invite to perform an investigation on a larger sample to verify if this result could be significant. In this case, in fact, the Candida spp presence in the PC group could be an element which, combined with some specific bacteria, could be a marker of pancreatic cancer.”

[63] Huët MAL, Lee CZ, Rahman S. A review on association of fungi with the development and progression of carcinogenesis in the human body. Current Research in Microbial Sciences. 2022 ;3:100090. DOI: 10.1016/j.crmicr.2021.100090. PMID: 34917994; PMCID: PMC8666644.

[64] Xu J, Zeng Y, Yu C, Xu S, Tang L, Zeng X, Huang Y, Sun Z, Xu B, Yu T. Visualization of the relationship between fungi and cancer from the perspective of bibliometric analysis. Heliyon. 2023 Jul 21;9(8):e18592. doi: 10.1016/j.heliyon.2023.e18592. PMID: 37529342; PMCID: PMC10388209.

[Reviewer]: Please check that all family, genus and species names are in italics.

[Reply]: We done

[Reviewer]: Line 161: replace "candida" with Candida".

[Reply]: We done 

[Reviewer]: English needs to be revised in some points: lines 56, 171, 177, 185, 219, 227, 244.

[Reply]: English language check was performed by a native speaker

---

## [Editor Report · Decision Letter 1]

25 Oct 2023

Microbial profile in bile from Pancreatic and Extra-Pancreatic Biliary Tract Cancer

PONE-D-23-14947R1

Dear Dr. Serra,

We’re pleased to inform you that your manuscript has been judged scientifically suitable for publication and will be formally accepted for publication once it meets all outstanding technical requirements.

Kind regards,

Massimiliano Galdiero, M.D., Ph.D.

Academic Editor

PLOS ONE
---

## [Editor Report · Acceptance letter]

30 Oct 2023

PONE-D-23-14947R1 

Microbial profile in bile from Pancreatic and Extra-Pancreatic Biliary Tract Cancer 

Dear Dr. Serra:

I'm pleased to inform you that your manuscript has been deemed suitable for publication in PLOS ONE. Congratulations! Your manuscript is now with our production department. 

Kind regards, 

on behalf of

Prof. Massimiliano Galdiero 

Academic Editor

PLOS ONE